# Examination of the Functional Relationship between *PD-L1* DNA Methylation and mRNA Expression in Non-Small-Cell Lung Cancer

**DOI:** 10.3390/cancers15061909

**Published:** 2023-03-22

**Authors:** Trine V. Larsen, Nina Dybdal, Tina F. Daugaard, Johanne Lade-Keller, Lin Lin, Boe S. Sorensen, Anders L. Nielsen

**Affiliations:** 1Department of Biomedicine, Aarhus University, 8000 Aarhus, Denmark; tvl@biomed.au.dk (T.V.L.); 201704447@post.au.dk (N.D.); tfm@biomed.au.dk (T.F.D.); lin.lin@biomed.au.dk (L.L.); 2Department of Pathology, Aarhus University Hospital, 8200 Aarhus, Denmark; johanne.keller@aarhus.rm.dk; 3Department of Clinical Biochemistry, Aarhus University Hospital, 8200 Aarhus, Denmark; boesoere@rm.dk; 4Department of Clinical Medicine, Aarhus University, 8000 Aarhus, Denmark

**Keywords:** immunotherapy, lung cancer, PD-L1, epigenetics, DNA methylation, biomarker

## Abstract

**Simple Summary:**

For non-small-cell lung cancer (NSCLC) patients undergoing immunotherapy blocking the PD-1/PD-L1 interaction, the cancer cell PD-L1 expression level is a determinant of treatment efficiency. This study was conducted to determine whether *PD-L1* DNA methylation regulates and can predict PD-L1 expression in NSCLC. Tumor biopsies and cell lines were analyzed for *PD-L1* DNA methylation, mRNA expression, and protein expression. CRISPR-Cas9 and dCas9 fusions with TET1 and DNMT3A were used to change the *PD-L1* DNA methylation levels. In this study, we found that the *PD-L1* DNA methylation status functionally influences its expression. However, although methylation can be inversely correlated with the expression of PD-L1 in NSCLC lines, this association is not strong in NSCLC tumor samples. The fact that *PD-L1* DNA methylation status does not inevitably mirror the expression level is important for future attempts to improve the effectiveness of PD-1/PD-L1 immunotherapy in NSCLC.

**Abstract:**

Immunotherapy targeting the interaction between programmed cell death protein 1 (PD-1) and programmed death-ligand 1 (PD-L1) is a treatment option for patients with non-small-cell lung cancer (NSCLC). The expression of PD-L1 by the NSCLC cells determines treatment effectiveness, but the relationship between *PD-L1* DNA methylation and expression has not been clearly described. We investigated *PD-L1* DNA methylation, mRNA expression, and protein expression in NSCLC cell lines and tumor biopsies. We used clustered regularly interspaced short palindromic repeats-associated protein 9 (CRISPR-Cas9) to modify *PD-L1* genetic contexts and endonuclease deficient Cas9 (dCas9) fusions with ten-eleven translocation methylcytosine dioxygenase 1 (TET1) and DNA (cytosine-5)-methyltransferase 3A (DNMT3A) to manipulate *PD-L1* DNA methylation. In NSCLC cell lines, we identified specific *PD-L1* CpG sites with methylation levels inversely correlated with *PD-L1* mRNA expression. However, inducing *PD-L1* mRNA expression with interferon-γ did not decrease the methylation level for these CpG sites, and using CRISPR-Cas9, we found that the CpG sites did not directly confer a negative regulation. dCas9-TET1 and dCas9-DNMT3A could induce *PD-L1* hypo- and hyper-methylation, respectively, with the latter conferring a decrease in expression showing the functional impact of methylation. In NSCLC biopsies, the inverse correlation between the methylation and expression of PD-L1 was weak. We conclude that there is a regulatory link between *PD-L1* DNA methylation and expression. However, since these measures are weakly associated, this study highlights the need for further research before *PD-L1* DNA methylation can be implemented as a biomarker and drug target for measures to improve the effectiveness of PD-1/PD-L1 immunotherapy in NSCLC.

## 1. Introduction

Programmed cell death protein 1 (PD-1) is a co-inhibitory surface receptor expressed on a wide range of immune cells, including antigen-stimulated T cells [1,2]. Programmed death-ligands 1 and 2 (PD-L1 and PD-L2) are ligands for PD-1, and the PD-1/PD-ligand axis is involved in maintaining peripheral and central immune cell tolerance [3]. Cancer and stromal cells can express PD-L1 [4]. PD-L1 expression can be induced by oncogenic driver mutations, e.g., epidermal growth factor receptor (*EGFR*) and Kirsten rat sarcoma virus (*KRAS*) mutations and anaplastic lymphoma kinase (*ALK*) chromosomal rearrangements [5,6,7,8]. Several immune-cell-produced pro-inflammatory molecules, such as the type I and II interferons (IFNs), IFN-α/β, and IFN-γ, also induce cancer cell PD-L1 expression [9]. Engagement of the PD-1 receptor with PD-L1 alters T cell activity, including the inhibition of T cell proliferation, survival, and cytokine production, as well as other T cell effector functions [4]. Thus, PD-L1 expression by cancer cells creates an immunosuppressive tumor microenvironment beneficial for the proliferation and survival of tumor cells.

Non-small-cell lung cancer (NSCLC) is a heterogeneous disease and, in approximately 20% of cases, characterized by oncogenic driver mutations in, e.g., *EGFR*, *KRAS*, and *ALK* [10]. Chemotherapy and target therapies addressing specific oncogenic drivers are treatment options. The latter is exemplified by EGFR-targeted therapy using several generations of tyrosine kinase inhibitors (TKI) to achieve control of NSCLC progression [11]. Despite an initial clinical significance, treatment resistance inevitably arises. EGFR TKI resistance mechanisms are pleiotropic and involve secondary *EGFR* mutations, amplification of the mesenchymal–epithelial transition factor proto-oncogene (*c-MET*), and epithelial–mesenchymal transition (EMT) [12]. Immunotherapy blocking PD-1 interaction with PD-L1 has changed the treatment pattern of NSCLC [13,14]. PD-1/PD-L1 immunotherapy with nivolumab (anti-PD-1), pembrolizumab (anti-PD-1), and atezolizumab (anti-PD-L1) are approved by the USA Food and Drug Administration for the treatment of advanced NSCLC [15,16,17,18,19,20]. PD-1/PD-L1 immunotherapy resistance in NSCLC is described, e.g., as being attributed to the loss-of-function of IFN signaling and the resulting decrease in IFN-γ-induced PD-L1 expression in cancer cells [21]. The PD-L1 tumor proportion score (TPS) determined by immunohistochemistry (IHC) is used as a predictive biomarker for successful immunotherapy in NSCLC [18,19,20,22,23,24,25]. However, up to 15% of patients scoring with PD-L1-negative tumors will respond to immunotherapy, and many patients with tumors scoring with a positive PD-L1 TPS do not benefit from the treatment [14,20,26]. Given the lack of a secure PD-L1-expression-based clinical stratification of patients as responders versus non-responders, NSCLC patients can be offered PD-1/PD-L1 immunotherapy irrespective of their PD-L1 TPS.

DNA methylation, the addition of a methyl group to the fifth position of cytosine in cytosine-phosphate-guanine (CpG) sites, regulates transcription without altering the genomic sequence. DNA (cytosine-5)-methyltransferases (DNMTs) establish (DNMT1) and maintain (DNMT3A/3B) DNA methylation, whereas ten-eleven translocation methylcytosine dioxygenase (TET) proteins, together with DNA repair pathways, inhibit DNA methylation [27,28,29,30]. In addition, methylated cytosine can be converted into thymine through spontaneous deamination. Regions called CpG islands with higher CpG contents than expected exist in the genome, indicating a selective pressure for maintenance. CpG islands can be flanked by shore and shelve regions with more diluted CpG contents. CpG islands have transcriptional regulatory potential, particularly if they are located in promoter regions. In general, promoter CpG islands have a bimodal methylation pattern, with a hypo-methylated status being the most commonly observed. Hyper-methylated CpG islands are potentially associated with transcriptional silencing of the corresponding gene, as exemplified by developmentally expressed genes [31]. CpG island shore and shelve regions have more dynamic methylation ranges, and the methylation of these regions can often be associated with transcriptional fine-tuning of the underlying gene. Open-sea CpG sites, which are individual CpG sites located in intragenic and intergenic regions, are mostly methylated, which, e.g., prevents spurious transcriptional initiation from intragenic regions and represses the transcription of repetitive and transposable elements [32]. Moreover, genome stability is supported by a CpG-methylated state, and, accordingly, global hypo-methylation is a hallmark of genome instability associated with carcinogenesis [33]. In NSCLC, global hypo-methylation can result in oncogene activation, and in addition, gene-specific hyper-methylation can result in tumor-suppressor gene inactivation.

The methylation status of CpG sites in the *PD-L1* CpG island shores is inversely correlated with *PD-L1* expression in acute myeloid leukemia and colorectal, melanoma, bladder, and prostate cancer [34,35,36,37,38]. In the case of NSCLC, this inverse correlation has also been described [39,40,41,42]. Concerning specific-shore CpG sites, the methylation of cg15837913 and cg19724470 in the CpG island north and south shores, respectively, is inversely correlated with *PD-L1* mRNA expression [34,37,38,43,44]. The prognostic value of *PD-L1* methylation has shown conflicting results, as both high and low *PD-L1* promoter methylation have been associated with shorter overall survival [34,35,36,37]. Hypomethylating agents of the azacytidine (AZA) family are cytosine analogs that can be incorporated into DNA and, as a consequence of an irreversible interaction with DNMTs, this results in genome-wide demethylation [45]. AZA can upregulate *PD-L1* mRNA and protein expression, at least in some NSCLC cells [42]. NSCLC patients having developed resistance to anti-PD-1 therapy showed increased *PD-L1* methylation and reduced mRNA and protein expression in the recurred tumor tissue [39]. However, AZA treatment resulted in the demethylation of the *PD-L1* promoter, increased *PD-L1* mRNA expression, and induced sensitization to anti-PD-1 therapy in a mouse model of immunotherapy resistance [39]. The IFN regulatory factors (IRFs) IRF1 and IRF7 can bind the *PD-L1* upstream promoter and mediate transcriptional activation [9]. In NSCLC, *IRF1* and *IRF7* expression can be repressed by a mechanism involving DNA methylation, and the methylation status is inversely correlated with PD-L1 expression [21,42]. The restoration of IRF1 and IRF7 expression with AZA-analogous decitabine reactivated PD-L1 expression, and in a mouse model, a concomitant enhanced efficiency of anti-PD-1 therapy was observed [21]. In line with these results, clinical trials are testing the anti-PD-1 antibody pembrolizumab combined with demethylation agents to further examine the potential of sensitizing NSCLC patients to PD-1/PD-L1 axis immunotherapy [46].

In the ongoing process of improving PD-1/PD-L1 NSCLC immunotherapy, it will be beneficial to decipher the mechanisms regulating PD-L1 expression to identify new treatment options and biomarkers for the identification of actual immunotherapy responders among NSCLC patients. It is not well understood whether an increase in *PD-L1* CpG island shore methylation directly inhibits *PD-L1* expression or is a consequence of inhibited *PD-L1* expression. In this study, we examined the interplay between *PD-L1* DNA methylation and expression in NSCLC.

## 2. Material and Methods

### 2.1. Cell Culture

Details concerning the applied NSCLC cell lines, including the place of purchase, are described in Appendix A. HEK293T was purchased from the American Type Culture Collection (ATCC). Cell lines were grown in RPMI-1640, except for HEK293T, Calu-3, and A427, which were grown in Dulbecco’s Modified Eagle’s Medium supplemented with glutamine. For all cells, the culture medium was supplemented with 10% fetal calf serum, 1% penicillin-streptomycin, 10 mM HEPES, and 1 mM sodium pyruvate. The cells were grown at 37 °C with 5% CO_2_. Erlotinib TKI-resistant HCC827 cell clones (HCC827ER) were grown in the presence of 5 μM erlotinib [47]. For IFN-γ induction, the cells were seeded in triplicate at a cell density resulting in 50–80% confluent cells at the time of harvest. The cells were stimulated with the culture medium containing IFN-γ (10 ng/mL) (PeproTech, London, UK, 300-02) for 24 h, 48 h, and 72 h. For cell stimulation with IFN-γ (10 ng/mL) for 7 days, the cell culture medium was replaced with a new culture medium, +IFN-γ, every third day. The control cells received culture medium with an equivalent volume of PBS + 0.01% bovine serum albumin (Sigma-Aldrich, Saint Louis, MO, USA, A2153). For treatment with AZA HCC827 (100,000 cells/well), A427 (125,000 cells/well), and HEK293T (150,000–300,000 cells/well), the cells were seeded in triplicate. On the following day, the seeding cells were treated with 1 μM AZA (Sigma-Aldrich, A2385). The cell culture medium was replaced with a fresh medium containing AZA every day. The control cells received a culture medium with an equivalent volume of DMSO.

### 2.2. Lung Tissue from NSCLC Patients

DNA methylation and *PD-L1* mRNA expression were examined in tissues from lung adenocarcinoma tumors (*n* = 31) without *EGFR* mutations or *ALK* translocations or normal adjacent tissue (*n* = 12) removed by resection or lobectomy at Aarhus University Hospital during the treatment and diagnostics of the tumors. The material was formalin-fixed for a minimum of 24 h and a maximum of 72 h, and the selected sections were paraffin-embedded according to routine diagnostic procedures. After routine diagnostics had been performed and the material had been archived, paraffin blocks with obvious invasive tumor areas were selected based on the hematoxylin-and-eosin-stained slides that were created during the routine diagnostics. Immunohistochemistry was performed on tissue sections (3–4 μm) using the 22C3 PharmDx assay (Agilent, Santa Clara, CA, USA), according to the manufacturer’s instructions. The PD-L1 TPS was assessed by a trained pathologist as the proportion of neoplastic cells with complete or partly membranous staining. The tumors were subsequently categorized into three groups (PD-L1 TPS < 1%, 1–49%, and ≥50%). The use of anonymized formalin-fixed paraffin-embedded (FFPE) lung tissue sections with NSCLC was requested and approved based on Danish and EU ethical guidelines. The project was evaluated as a methodological (non-health-science-research) study by the local ethical committee and was therefore exempt from additional approval by the national ethical committee (235/2018).

### 2.3. RNA Expression Analyses

mRNA expression analyses were performed in silico using data extracted from the DepMap Portal (https://depmap.org/portal/interactive/) (accessed on 2 August 2022) for NSCLC cell lines. cDNA synthesis of the RNA from the NSCLC cell lines was performed with an iScript cDNA synthesis kit (Bio-Rad, Hercules, CA, USA, 170-8890). First-strand cDNA synthesis of RNA from the FFPE tumor tissue was performed using 1 μg RNA with a SuperScript II Reverse Transcriptase kit (Invitrogen, Carlsbad, CA, USA, 18064). RT-qPCR was performed using the LightCycler 480 platform (Roche, Basel, Switzerland). The primer sequences are available in the Appendix A. Analysis of data was conducted using the X_0_ method with *TBP* or *ACTB* mRNA expression as a reference gene [48]. 

### 2.4. DNA Methylation Analyses

Infinium MethylationEPIC 850K Array (Illumina, San Diego, CA, USA) analyses were performed for the HCC827 parental and erlotinib-resistant cell clones as previously described [49]. For bisulfite-pyrosequencing, genomic DNA was processed with the EpiTect Bisulfite kit (Qiagen, Venlo, Netherlands). The genomic regions were amplified using a PyroMark PCR kit and by pyrosequencing using a PyroMark Q24 station (Qiagen, Venlo, The Netherlands) (details in Appendix A). The data were analyzed with PyroMark Q24 Advanced software (Qiagen, Venlo, The Netherlands), with all samples PCR-amplified and pyrosequenced in duplicate at the least. 

### 2.5. CRISPR-Cas9-Mediated Deletions

A427 and HCC827 cells with a stable expression of clustered regularly interspaced short palindromic repeats-associated protein 9 (Cas9) were generated using lentiviral transduction with pLentiCas9-blast (Addgene, Watertown, MA, USA, 52962), as previously described [50]. Short guide RNAs (sgRNA) were designed using the UCSC Genome Browser (Appendix A includes the sgRNA sequences). Annealed oligonucleotides containing the sgRNA sequence were cloned into the pLentiGuide-puro vector (Addgene, 52963) or the pLentiGuide-hygro vector (Addgene, 139462). A pLentiGuide-pyro vector expressing scrambled sgRNA was used as a control. After 14 days of blasticidin selection, the HCC827-Cas9 and A427-Cas9 cells were transduced with sgRNA expressing lentivirus. After 14 days of antibiotic selection, DNA was extracted. The genomic region was amplified, Sanger-sequenced, and analyzed with Synthego Inference of CRISPR Edits software (https://labs.synthego.com/) (accessed on 3 January 2022 (A427 cells) and 7 January 2022 (HCC827 cells)).

### 2.6. dSaCas9-TET1 and dSpCas9-DNMT3A Fusions

Doxycycline-inducible TRE3G expression cassettes [51,52] encoding Streptococcus aureus (Sa) fusions with endonuclease-deficient Cas9 (dCas9) [53], together with TET1 (PB-TRE-dSaCas9-TET1-Puro and PB-TRE-dSaCas9-dTET1-Puro), and Streptococcus pyogenes (Sp) fusions with dCas9 [54], together with DNMT3A (PB-TRE-dSpCas9-DNMT3A-Hygro and PB-TRE-dSpCas9-dDNMT3A-Hygro) were generated in this study. These fusion proteins are composed of the inactive dSaCas9 or dSpCas9 protein (D10A and N580A or D10A and H840A, respectively) C-terminally fused with the active catalytic domains of TET1 or DNMT3A or the corresponding inactive catalytic domains denoted as dTET1 (H1672Y and D1674A) or dDNMT3A (E752A) [55,56]. The expression cassettes were stably introduced into HEK293T cells using the piggyBac transposon system [57]. sgRNAs (Appendix A) were cloned into Lenti-sgRNA(SaCas9)-eGFP-Puro or Lenti-sgRNA(SpCas9)-eGFP-Puro. Expression of the dCas9 fusions was induced by 1 μg/mL doxycycline hyclate (Dox) (Sigma-Aldrich, Saint Louis, MO, USA, D9891). For the protein expression analysis, the cells were treated with Dox for 24 h before cell harvest for Western blot. For the targeted editing of DNA methylation, the cells were treated with Dox for 0.5 h–1 h before lipofectamine transfection using 2 μg sgRNA vector/well and cultured for 48 h. The Dox-containing medium was renewed after 1 day.

### 2.7. Statistical Analyses

Statistical analyses were performed using GraphPad Prism version 9. Data concerning the cell lines are presented as the mean with error bars as the standard deviation. Data concerning the NSCLC patients are presented as the median. The differences in means between groups were analyzed using the unpaired *t*-test and one-way ANOVA, and differences in medians between groups were analyzed using the Mann–Whitney *t*-test and the Kruskal–Wallis test. Correlation analysis was performed using Spearman’s correlation coefficient analysis. Data were considered significant when the *p*-value/adjusted *p*-value < 0.05. 

## 3. Results

### 3.1. Inverse Correlation between PD-L1 Methylation and mRNA Expression in NSCLC Cell Lines

Cancer-based studies have described an inverse correlation between *PD-L1* mRNA expression and methylation of cg19724470 in the CpG island south shore, situated 434 bp downstream of the *PD-L1* transcriptional start site (Appendix A) [34,35,38,58]. We envisaged exploring this association between cg19724470 DNA methylation and *PD-L1* mRNA expression in NSCLC cells. For this purpose, we used bisulfite-pyrosequencing. Most existing *PD-L1* methylation data are affected by constraints related to the implementation of the array-based methodology, which does not account for potentially informative CpG sites in the vicinity of cg19724470. To examine whether a CpG site located near cg19724470 also displays an association with *PD-L1* expression, we included a methylation analysis of an as-yet unannotated CpG site located 15 bases downstream of cg19724470 in the genomic position chr9:5,450,951 (hereafter abbreviated as cg(chr9:5450951)). To mimic the oncogenic spectrum observed in NSCLC patients, we analyzed 14 NSCLC cell lines with different oncogenic mutations, including hotspot mutations in *EGFR*, *KRAS*, tumor protein P53 (*TP53*)*,* and phosphatidylinositol 4,5-bisphosphate 3-kinase catalytic subunit alpha isoform (*PIK3CA*). cg19724470 and cg(chr9:5450951) displayed variable DNA methylation levels in the NSCLC cell lines (Figure 1A). Furthermore, the methylation of cg19724470 was strongly correlated with the methylation of cg(chr9:5450951) (Spearman *r* = 0.91, *p* < 0.0001) (Appendix A). *PD-L1* mRNA expression in the NSCLC cell lines was first calculated using data from the DepMap Portal. The NSCLC cell lines displayed *PD-L1* mRNA expression to different degrees (Figure 1B). *PD-L1* mRNA expression was inversely correlated with the DNA methylation of cg19724470 (Spearman *r* = −0.88, *p* = 0.0002) and cg(chr9:5450951) (Spearman *r* = −0.78, *p* = 0.0025) (Figure 1C). The validation of *PD-L1* mRNA expression by RT-qPCR showed a similar trend of inverse correlation between *PD-L1* mRNA expression and DNA methylation (Appendix A). Altogether, the analyses showed that across the NSCLC cell lines, the DNA methylation levels of both cg19724470 and cg(chr9:5450951) have an inverse correlation with *PD-L1* mRNA expression.

To address the eventual existence of additional *PD-L1* CpG sites with methylation levels correlating with the mRNA expression observed in NSCLC, we screened a previously generated dataset including genome-wide information on DNA methylation (Infinium MethylationEPIC 850K array) and mRNA expression (RNA-sequencing) from a group of five isogenic HCC827 cell clones [49]. These included parental HCC827 cells, as well as erlotinib TKI-resistant cell clones, with the amplification of c-*MET* (HCC827ER2 and HCC827ER3) and EMT (HCC827ER4 and HCC827ER10) [47]. HCC827ER2 and HCC827ER3 have higher *PD-L1* mRNA expression relative to parental HCC827 cells, whereas HCC827ER4 and HCC827ER10 have lower *PD-L1* mRNA expression (Appendix A) [59]. Across the HCC827 cell clones, the *PD-L1* CpG sites in the CpG island in the promoter region were unmethylated, whereas CpG sites located within the south shore corresponding to the gene body were hypermethylated (Appendix A). Notably, we did not observe the differential methylation of cg19724470 among the HCC827 cell clones, whereas, for cg(chr9:5450951), a tendency of inverse correlation between methylation and *PD-L1* mRNA expression was noted (Appendix A). In the region corresponding to the 3′-end of intron 1, three of the four CpG sites showed methylation levels displaying variability (ranging from approximately 10% to 50% methylation) (Appendix A). However, a correlation between *PD-L1* mRNA expression and the methylation levels was not apparent (Appendix A). To substantiate this negative result, we further explored two of these CpG sites, cg04478497 and cg00975815, located at 4561 bp and 4569 bp, respectively, downstream of the *PD-L1* transcriptional start site. We designed a bisulfite-pyrosequencing assay for methylation quantification covering both sites. For neither the HCC827 cell clones nor a group of 13 additional NSCLC cell lines did we identify a significant correlation between the DNA methylation of these CpG sites and mRNA expression (Appendix A). Altogether, the above analyses did not reveal the presence of CpG sites that presented DNA methylation levels correlating with a *PD-L1* mRNA expression that was more pronounced than that observed for cg19724470 and cg(chr9:5450951).

### 3.2. IFN-γ-Mediated Induction of PD-L1 mRNA Expression Does Not Drive Concomitant DNA Methylation Changes

The tumor milieu operates through the production of cytokines and IFNs, e.g., T-cell-produced IFN-γ, determining the PD-L1 expression level, in addition to the effects of oncogenic signaling pathways [9,60,61]. To further examine the association between *PD-L1* mRNA expression and DNA methylation in NSCLC cells, we examined the impact of IFN-γ. Stimulation with IFN-γ for 48 h and 72 h induced *PD-L1* mRNA expression in A427 and HCC827 cells (Figure 2A,B). Short-term stimulation (24 h) induced *PD-L1* mRNA expression in HCC827 cells (Figure 2B). However, stimulation with IFN-γ for these periods had no significant effect on methylation for either cg19724470 and cg(chr9:5450951) or cg04478497 and cg00975815 (Figure 2A,B and Appendix A). Induction with IFN-γ for a longer period could, in principle, be required for changes in DNA methylation to manifest. Thus, we performed stimulation with IFN-γ for 7 days. This resulted in increased *PD-L1* mRNA expression in both the A427 and HCC827 cells (Figure 2A,B). However, even the long-term stimulation with IFN-γ did not influence the DNA methylation levels (Figure 2A,B). Altogether, the results indicate that *PD-L1* CpG island south shore methylation levels do not mirror the *PD-L1* mRNA expression induced upon IFN-γ stimulation. Thus, PD-L1 methylation is not an invariable corresponding to mRNA expression. In an immunotherapy context, the results could support the notion that, for a tumor in which an increase in cancer cell PD-L1 expression is mediated through T-cell-secreted IFN-γ, the *PD-L1* methylation level will not be a marker of actual PD-L1 expression.

### 3.3. The Differentially Methylated Region including cg19724470 and cg(chr9:5450951) Does Not Actively Regulate PD-L1 mRNA Expression

With the above results showing that the methylation status of cg19724470 and cg(chr9:5450951) is inversely correlated with *PD-L1* mRNA expression in the absence of IFN-γ, we questioned whether the particular part of the *PD-L1* gene harboring these CpG sites and its methylation status are directly involved in the regulation of expression. We note that this region of the *PD-L1* gene contains no characterized IFN response elements or other described cis-elements. To create *PD-L1* deletions, we utilized CRISPR-Cas9 to generate indels in A427 cells with relatively high DNA methylation and low *PD-L1* mRNA expression and in HCC827 cells with relatively low DNA methylation and high *PD-L1* mRNA expression. Following the testing of various single and combined sgRNAs, we found a pair of sgRNAs resulting in a high fraction of cells harboring a deletion of both cg19724470 and cg(chr9:5450951) (Figure 3A and Appendix A). The mRNA expression analysis of the effect of these deletions showed a significant but negligible upregulation of *PD-L1* mRNA expression in the HCC827 cells and no significant alteration in *PD-L1* mRNA expression in the A427 cells (Figure 3B and Appendix A). We conclude that the genomic region containing cg19724470 and cg(chr9:5450951), presenting either with or without DNA methylation, does not play an important, direct role in the regulation of *PD-L1* mRNA expression.

The treatment of cells with AZA results in genome-wide demethylation and, accordingly, is used one approach used to determine whether decreased *PD-L1* DNA methylation results in increased mRNA expression. PD-L1 expression was previously shown to respond diversely to AZA treatment in different NSCLC cells [42]. In line with the expected genome-wide decreasing effect on DNA methylation levels mediated by AZA, we observed a reduced methylation level for the examined *PD-L1* CpG sites in A427 cells (Figure 3C and Appendix A). However, *PD-L1* mRNA expression was not concomitantly increased (Figure 3C). HCC827 possesses a low baseline *PD-L1* DNA methylation level, particularly in the case of the cg197724470 and cg(chr9:5450951) of the examined CpG sites, eventually complicating further demethylation (Figure 3D and Appendix A). Indeed, AZA had no significant effect on cg197724470 and cg(chr9:5450951) methylation, whereas some demethylation was observed for cg04478497 and cg00975815 (Figure 3D and Appendix A). No increase in *PD-L1* mRNA expression was observed following treatment of the HCC827 cells with AZA; instead, a minor decrease in expression was observed (Figure 3D). In conclusion, the AZA experiments did not support the notion that a genome-wide-mediated decrease in DNA methylation, which also includes minor decreases in methylation for *PD-L1*, is sufficient to confer an increase in *PD-L1* mRNA expression. 

### 3.4. Targeted Methylation with dSpCas9-DNMT3A Decreases PD-L1 mRNA Expression

As AZA is a non-specific drug, it is not clear whether the above-observed lack of effect on *PD-L1* expression is an indirect outcome of genome-wide demethylation. To explore the impact of direct recruitment of enzymatic activities involved in methylation and demethylation specifically in regard to the *PD-L1* gene, we examined a series of fusions with enzymatically dead Sa and Sp Cas9 (dSaCas9 and dSpCas9, respectively). First, we examined the catalytic domain of DNA demethylase TET1 fused to dSaCas9 (dSaCas9-TET1) in the Dox-inducible HEK293T cell model. For the control, we also included dSaCas9 fused to a catalytically dead TET1 domain (H1672Y and D1674A) (dSaCas9-dTET1). The HEK293T cell model was selected since this non-cancer cellular background could enable the delineation of the impact of methylation on *PD-L1* mRNA expression without interference from specific oncogenic driver mutations and pathways activated in the collection of the NSCLC cell lines. HEK293T cells resemble A427 cells in that they possess low *PD-L1* mRNA expression and high baseline DNA methylation, an AZA-resistant *PD-L1* mRNA expression level, and no change in *PD-L1* methylation following IFN-γ-mediated induction of mRNA expression. We validated the Dox-inducible expression of dSaCas9 fusions by Western blotting (Appendix A). dSaCas9-TET1 cells were transiently transfected with expression vectors for with a pool of five Sa sgRNAs (Sa sgRNA pool) targeting the genomic regions upstream and downstream of cg19724470 and cg(chr9:5450951). cg19724470 and cg(chr9:5450951) were significantly demethylated 48 h after transfection with the sgRNAs (Figure 4A). We note that the methylation assay does not discriminate between 5-methylcytosine and 5-hydroxymethylcytosine and, accordingly, the determined methylation level is potentially overestimated. In the dSaCas9-dTET1 cells, demethylation was not observed (Appendix A). *PD-L1* mRNA expression could be impeded by the tethering of the dSaCas9 fusions with the DNA, independent of a demethylation effect of dSaCas9-TET1 [62]. Adjustment for this factor by comparing the effect of dSaCas9-TET1 relative to dSaCas9-dTET1 revealed a non-significant trend of increased *PD-L1* mRNA expression (Figure 4B). We conclude that inducing a gene-specific decrease in *PD-L1* methylation is insufficient to drive a significant increase in mRNA expression, which is in line with the above observations following AZA treatment. 

Next, we determined whether a gene-specific increase in *PD-L1* methylation could result in a decrease in expression. For this purpose, we used Sp dCas9 fused with the catalytic domain of methyltransferase DNMT3A (dSpCas9-DNMT3A) or as a control fused with a catalytically inactive DNMT3A domain as a result of the E752A mutation (dSpCas9-dDNMT3A). We validated the Dox-inducible expression of the dCas9 fusions by Western blotting (Appendix A). The transfection of the HEK293T dSpCas9-DNMT3A cells with a pool of Sp sgRNAs targeting the north and south shores of the *PD-L1* CpG island (Sp sgRNA pool) significantly increased cg19724470 but not cg(chr9:5450951) methylation (Figure 4C). We note the high baseline methylation status of these two CpGs, eventually complicating the conferral of further methylation (Figure 4C). An additional analysis of the 3′ end of the *PD-L1* CpG island and its associated shore region further supported the fact that there was a gain in targeted *PD-L1* methylation due to the presence of the Sp sgRNA pool and that this methylation spread into the otherwise unmethylated CpG island (Figure 4D and Appendix A). Adjustment for the eventual presence of *PD-L1* transcriptional blocking from the genome tethering of the dSpCas9-fusions revealed that the recruitment of the catalytically active DNMT3A domain decreased *PD-L1* mRNA expression (Figure 4E). Thus, increasing the methylation status of the *PD-L1* gene corresponding to the region containing the CpG island at the promoter can directly decrease *PD-L1* mRNA expression. Hence, this supports the notion that *PD-L1* DNA methylation can functionally confer a decrease in mRNA expression.

### 3.5. Methylation of cg19724470 Is Correlated with PD-L1 TPS in NSCLC Patients

Finally, we examined whether there was an inverse correlation between *PD-L1* methylation and mRNA expression in NSCLC tumor biopsies. DNA methylation was determined by pyrosequencing, and mRNA expression was determined by RT-qPCR. The DNA methylation of cg19724470 and cg(chr9:5450951) was not significantly correlated with *PD-L1* mRNA expression, although there was a trend of inverse correlation between cg19724470 and *PD-L1* mRNA expression (spearman *r* = −0.33, *p* = 0.0906) (Figure 5A). Despite the fact that the cg04478497 and cg00975815 methylation levels were not correlated with *PD-L1* mRNA expression in the NSCLC cell lines, we questioned whether a correlation was present in the NSCLC tumor tissue. However, no significant correlations were present (Appendix A). Next, we revealed PD-L1 protein expression in the tumor biopsies based on immunohistochemistry (Appendix A), and the PD-L1 TPS was determined. A significant inverse correlation was observed between the methylation of cg19724470 and PD-L1 TPS (Figure 5B). This was not observed for the other examined CpG sites (Figure 5B and Appendix A). Division of the tumors into groups according to their PD-L1 TPS (<1%, 1–49%, and ≥50%) did not result in a significant difference between the methylation medians of the groups for any of the examined CpG sites (Appendix A). However, a trend of higher and more diverse methylation of cg19724470 was observed in the PD-L1 TPS < 1% group compared to the 1–49% and ≥50% groups (Appendix A). A similar trend was observed for cg(chr9:5450951) (Appendix A). We note the expected positive correlation between *PD-L1* mRNA expression and PD-L1 TPS (*r* = 0.64, *p* = 0.0003) (Appendix A). In conclusion, we only observed a negligible inverse correlation between *PD-L1* methylation and expression, which indicates that in samples representing a complex tumor milieu harboring various cell types and cytokines/IFNs, the *PD-L1* methylation level does not strongly inverse correlate with the PD-L1 expression level. 

We also examined the relation between *PD-L1* methylation and mRNA expression in tumor tissues relative to adjacent normal lung tissues. We observed no significant differences in the median methylation of any of the four examined CpG sites (Appendix A), nor did we identify a significant difference in median *PD-L1* mRNA expression between the normal adjacent tissues and tumor tissues (Appendix A). A greater variation in *PD-L1* mRNA expression was present in the tumor relative to the normal tissues (Appendix A). Examination of the paired normal and tumor tissues for *PD-L1* methylation and mRNA expression revealed both increases and decreases, highlighting differences between the tumor and normal tissues that are not systematically present in NSCLC tumor samples (Figure 5C and Appendix A). We conclude that in the analyzed sample set, neither *PD-L1* DNA methylation nor mRNA expression significantly differs between paired normal and tumor tissues.

## 4. Discussion

Here, we describe different aspects of the interplay between *PD-L1* DNA methylation and mRNA expression. We identified that in NSCLC cells, the methylation of cg19724470, located in the south shore of the CpG island, is inversely correlated with *PD-L1* mRNA expression. We also identified an inverse correlation for cg(chr9:5450951), located downstream of cg19724470. This is in line with other studies showing an inverse relationship between *PD-L1* DNA methylation and mRNA expression [34,35,36,37,38,39,40,41,42,43,44]. The inverse correlation between *PD-L1* mRNA expression and DNA methylation could reflect an active function of the methylation of these positions in driving the transcriptional repression of *PD-L1* and/or the fact that methylation is a passenger event for a given *PD-L1* expression level. Deletion of the *PD-L1* gene sequences containing cg19724470 and cg(chr9:5450951) did not alter *PD-L1* mRNA expression in the A427 and HCC827 cells. This indicates that this region of the *PD-L1* gene does not include important transcriptional regulatory *cis*-elements whose DNA methylation status directly regulates *PD-L1* expression. Moreover, the growth of the cells in the presence of the DNA-demethylating reagent AZA did not, per se, induce *PD-L1* mRNA expression. Altogether, these results lead us to question the impact of *PD-L1* methylation in driving the repression of *PD-L1* mRNA expression. The best direct evidence for an at least modest impact of *PD-L1* methylation status in driving mRNA expression was derived from the dCas9 fusion analyses, showing that gene-targeted methylation with DNMT3A decreased *PD-L1* expression, whereas gene-targeted demethylation with TET1 showed a tendency towards increased expression. In light of this, it is important to note that these data were gathered from a cell line with a genetic background absent of oncogenic driver mutations, HEK293T, whereas the NSCLC cell lines HCC827 and A4427 harbor activating oncogenic driver mutations in *EGFR* and *KRAS*, respectively. In NSCLC, *EGFR* and *KRAS* mutations regulate *PD-L1* expression through different cell signal transduction pathways [6,63,64]. Thus, despite the fact that our analyses do not support the notion that *PD-L1* methylation plays a major role in dictating *PD-L1* expression, it might have a regulatory function in concert with specific cancer-cell-intrinsic signaling pathways or external stimuli generated in the tumor microenvironment, i.e., pro-inflammatory cytokines, which are known inducers of *PD-L1* mRNA expression. A study conducted by Asgarova et al. supports this argument [41]. In *KRAS*-mutated NSCLC cells, the addition of transforming growth factor-β1 (TGF-β1) decreased DNMT1 activity, with resulting *PD-L1* promoter hypo-methylation. The addition of tumor necrosis factor-α (TNF-α) activated nuclear factor kappa B (NFκB) signaling. Neither TGF-β1 nor TNF-α could, by themselves, activate *PD-L1* expression [41]. However, when added simultaneously, the TGF-β1-mediated *PD-L1* promoter hypo-methylation allowed for the recruitment of the TNF-α-activated NFκB transcriptional activator complex, resulting in increased *PD-L1* expression [41]. Along the same lines, Wrangle et al. showed that AZA, in general, has a minor impact on *PD-L1* mRNA expression in NSCLC cell lines, but at the same time, the effect size depended on the precise oncogenic signal pathways activated in the given NSCLC cells [42]. 

IFNs, e.g., IFN-γ, secreted by cytotoxic T cells in the tumor microenvironment, can induce cancer cell PD-L1 expression [9]. In this study, as expected, we found that stimulating NSCLC cell lines with IFN-γ induced *PD-L1* mRNA expression. However, IFN-γ stimulation, even when implanted for up to seven days, did not affect *PD-L1* methylation. Thus, the external induction of *PD-L1* expression in cancer cells, e.g., in the tumor microenvironment, may have the ability to bypass concordant changes in *PD-L1* promoter methylation. Furthermore, we found that in individual patient tumor samples, the *PD-L1* mRNA expression levels were not invariably associated with concordant DNA methylation levels, further indicating the significance of factors beyond *PD-L1* methylation in regulating *PD-L1* expression. NSCLC tumors with 1–49% and ≥50% PD-L1-positive tumor cells did not display *PD-L1* methylation patterns which could explain their different degrees of PD-L1 expression. The methylation level of cg19724470 in the NSCLC tumors more closely resembled an on-off switch for PD-L1 expression, as cg19724470 methylation of the tumors with PD-L1 TPS ≥ 1%, in general, was lower compared to the methylation observed in tumors with PD-L1 TPS < 1%. Thus, the difference in PD-L1 expression between tumors with PD-L1 TPS ≥ 1% must be explained by factors other than the methylation of CpG sites, at least in the case of those examined here. Notably, PD-L1 is also regulated on the post-transcriptional and post-translational levels, as exemplified by miRNAs regulating *PD-L1* mRNA stability; chemokine-like-factor (CKLF)-like MARVEL transmembrane domain-containing proteins 4 and 6 (CMTM4 and CMTM6), regulating PD-L1 protein translocation at the cancer cell membrane; and various modifications regulating PD-L1 localization and stability [65,66,67,68]. However, one should not ignore the fact that the regulation of *PD-L1* on the transcriptional level can impact the amount of cell-surface-located PD-L1 [8,69]. In line with this, we found that PD-L1 TPS and *PD-L1* mRNA expression were significantly correlated in NSCLC tumors (Spearman’s *r* = 0.64, *p* = 0.0003).

## 5. Conclusions

The presented results highlight that in NSCLC cells, a negative correlation can exist between *PD-L1* DNA methylation and expression and that a gain in *PD-L1* DNA methylation can be functionally involved in mediating a decrease in expression. However, the results also show that in the context of NSCLC tumors, the negative correlation between *PD-L1* DNA methylation and expression is weak, and this could reflect the fact that, e.g., the IFN-γ-mediated induction of *PD-L1* expression does not result in DNA methylation level changes. Since *PD-L1* DNA methylation and mRNA expression levels appear to be only weakly associated, our findings highlight the need for further research before *PD-L1* DNA methylation can be implemented as a biomarker and drug target for measures to improve the effectiveness of PD-1/PD-L1 immunotherapy in NSCLC.

## Figures and Tables

**Figure 1 cancers-15-01909-f001:**
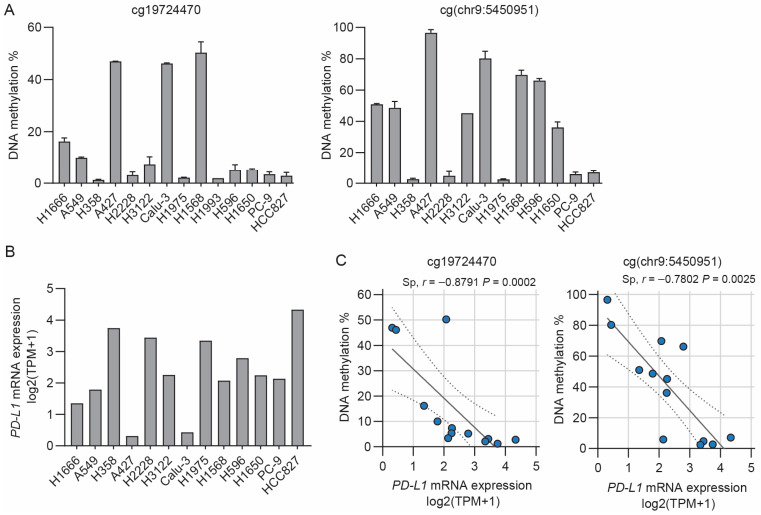
*PD-L1* methylation and mRNA expression in NSCLC cell lines. (**A**) DNA methylation of cg19724470 (left panel) and cg(chr9:5450951) (right panel) in NSCLC cell lines. Mean ± SD of at least two PCR-pyrosequencing runs. For cg(chr9:5450951), H1993 is not included, as we could not successfully quantify the DNA methylation at this site. (**B**) *PD-L1* mRNA expression in NSCLC cell lines from the Expression 22Q2 Public dataset derived from DepMap Portal. The DepMap Portal contained no information on the H1993 cell line at the time of the analyses. Transcripts per million (TPM). (**C**) Correlation analysis of *PD-L1* mRNA expression from the DepMap Portal and DNA methylation of cg19724470 (left panel) and cg(chr9:5450951) (right panel) in NSCLC cell lines. Spearman’s correlation coefficients (*r*) and the *p*-values are depicted above the graphs. Linear best fit (solid line) and 95% confidence intervals (dotted lines) are depicted.

**Figure 2 cancers-15-01909-f002:**
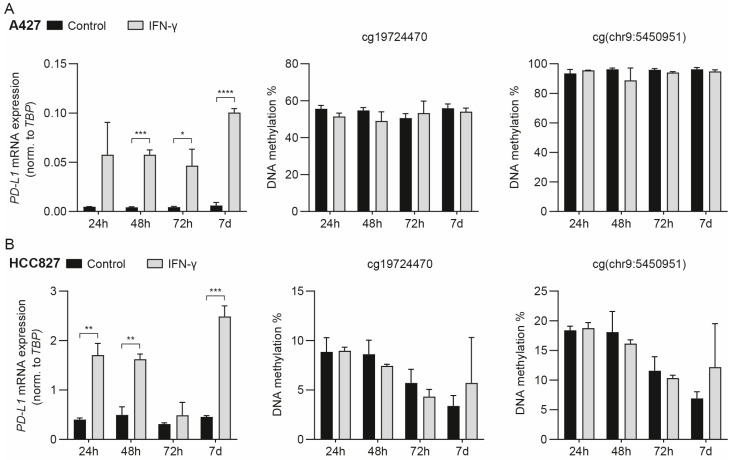
*PD-L1* methylation and mRNA expression following interferon-γ (IFN-γ) induction. (**A**) A427 cells and (**B**) HCC827 cells were stimulated with 10 ng/mL IFN-γ for 24 h, 48 h, 72 h, or 7 days. *PD-L1* mRNA expression was normalized to *TBP*. DNA methylation of cg19724470 and cg(chr9:5450951). Mean ± SD. *n* = 3 (except for the A427 control and IFN-γ at 24 h, where *n* = 2). Differences between the control and IFN-γ treatment were tested through an unpaired *t*-test with correction for multiple comparisons using the Holm–Šidák method. Adjusted *p*-value * < 0.05; ** < 0.01; *** < 0.001; **** < 0.0001.

**Figure 3 cancers-15-01909-f003:**
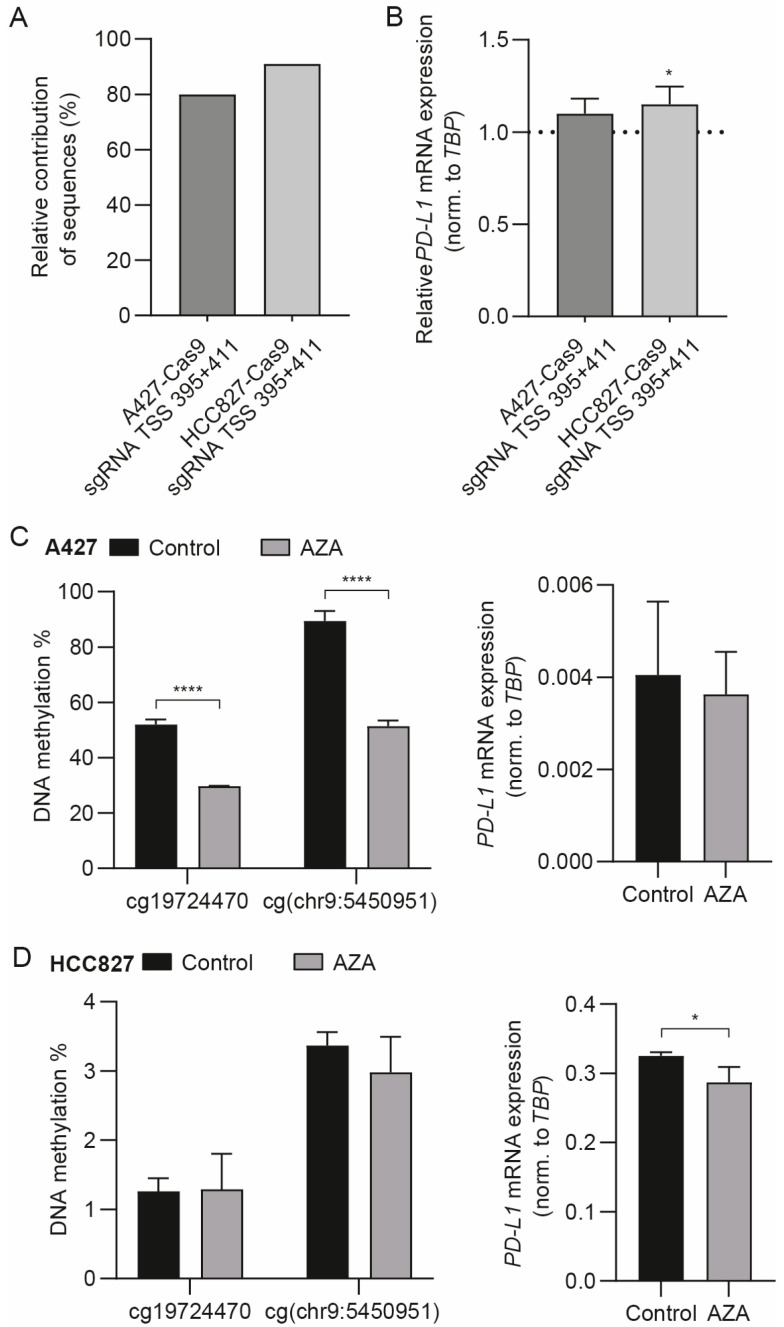
*PD-L1* mRNA expression following alteration in genome-wide DNA methylation by azacytidine (AZA). (**A**,**B**) The consequence of a *PD-L1* deletion harboring cg19724470 and cg(chr9:5450951). (**A**) The relative contributions of sequences with the deletion of both cg19724470 and cg(chr9:5450951). (**B**) *PD-L1* mRNA expression normalized to *TBP* followed by normalization to cells transduced with the control sgRNA. The dotted line represents normalized expression with the control sgRNA. Mean ± SD of technical triplicates. (**C**,**D**) Effect of 48 h of AZA treatment on *PD-L1* mRNA expression and methylation in (**C**) A427 cells and (**D**) HCC827 cells. Mean ± SD of technical triplicates. Differences in *PD-L1* mRNA expression between the control and AZA treatment were tested through an unpaired *t*-test. Differences in DNA methylation between the control and AZA treatment were tested through an unpaired *t*-test with correction for multiple comparisons using the Holm–Šidák method. Ordinary one-way ANOVA with Holm–Šidák’s multiple comparisons test was used to test for differences in *PD-L1* expression. Adjusted *p*-value * < 0.05; **** < 0.0001.

**Figure 4 cancers-15-01909-f004:**
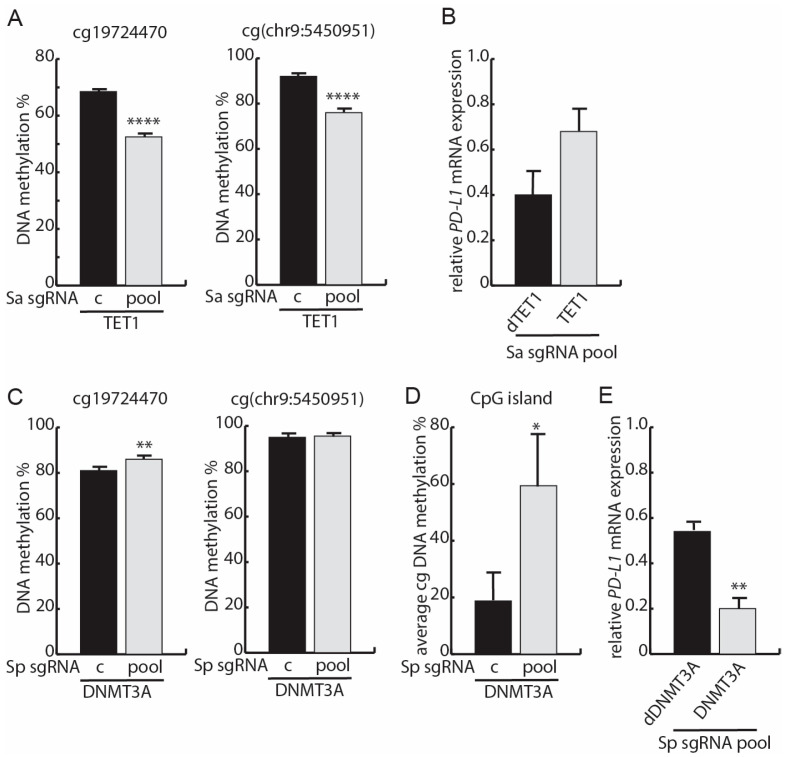
Effects of dCas9-fusion-mediated specific targeting of *PD-L1* methylation and demethylation. (**A**) HEK293T-dSaCas9-TET1 cells were induced with doxycycline and transfected with Streptococcus aureus (Sa) control sgRNA (c) or a pool of Sa sgRNAs (pool) targeting the south shore of the *PD-L1* CpG island. Next, 48 h after transfection, DNA methylation was analyzed and presented as mean ± SD. *n* = 3. One-way ANOVA with Holm–Šidák’s multiple comparisons test was used to test for significant differences in DNA methylation. (**B**) Relative *PD-L1* mRNA expression in dSaCas9-TET1 and dSaCas9-dTET1 cells transfected with the Sa sgRNA pool. The expression is shown relative to the values obtained with the Sp sgRNA control and normalized to *TBP* expression. An unpaired *t*-test was used to test for significance. (**C**) As in (**A**) but for HEK293T-dSpCas9-DNMT3A cells transfected with Streptococcus pyogenes (Sp) control sgRNA or a pool of Sp sgRNAs targeting the north and south shores of the *PD-L1* CpG island. *n* = 2. An unpaired *t*-test was used to test for significance. (**D**) Average CpG site methylation percentage calculated by clonal bisulfate sequencing of the 24 CpGs in the 3′ end of the CpG island and its associated shore in HEK293T dSpCas9-DNMT3A cells transfected with Sp control sgRNA or a pool of Sp sgRNAs targeting the north and south shores of the *PD-L1* CpG island. *n* = 4 sequences for each transfection. An unpaired *t*-test was used to examine for significance. *p*-value * < 0.01. (**E**) Relative *PD-L1* mRNA expression in dSpCas9-DNMT3A and dSpCas9-dDNMT3A cells transfected with the sgRNA pool. The expression is shown relative to the values obtained with the Sp sgRNA control and normalized to *TBP* expression. An unpaired *t*-test was used to examine for significance. *n* = 2 for DNMT3A and *n* = 3 for dDNMT3A. *p*-value or adjusted *p*-value ** < 0.01; **** < 0.0001.

**Figure 5 cancers-15-01909-f005:**
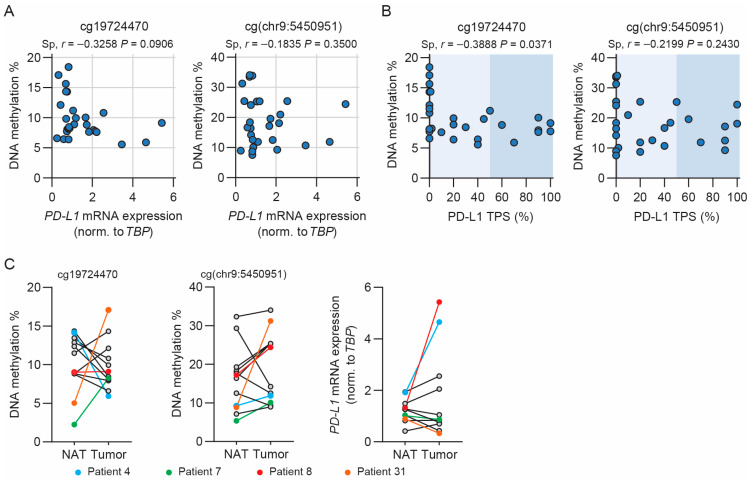
*PD-L1* methylation and mRNA expression in NSCLC tumor biopsies. (**A**) Spearman’s correlation analysis of *PD-L1* mRNA expression and DNA methylation of cg19724470 and cg(chr9:5450951), respectively, in NSCLC tumor tissue (*n* = 28). Spearman’s correlation coefficients (*r*) and the *p*-values are depicted above the graph. (**B**) Correlation analysis of PD-L1 TPS and DNA methylation of cg19724470 (*n* = 29) and cg(chr9:5450951) (*n* = 30), respectively, in NSCLC tumor tissue. Spearman’s correlation coefficients (*r*) and the *p*-values are depicted above the graph. Shading represents clinically used PD-L1 TPS subdivisions (<1%, 1–49%, and ≥50%). (**C**) DNA methylation of cg19724470 (*n* = 12) and cg(chr9:5450951) (*n* = 12) and *PD-L1* mRNA expression normalized to *TBP* (*n* = 11) in normal adjacent tissue (NAT) and tumor tissue, connected by a line for each patient. The selected patients are marked by color.

## Data Availability

All data will be shared upon reasonable request to the corresponding author.

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
