# Peer review of "Examination of the Functional Relationship between PD-L1 DNA Methylation and mRNA Expression in Non-Small-Cell Lung Cancer"

_cancers, 2023, doi:10.3390/cancers15061909_

Round 1

Reviewer 1 Report

Drs. Larsen and colleagues report on the relationship of methylation and mRNA expression of PDL1 in NSCLC. They have compared methylation and mRNA expression in different lung carcinoma cell lines and also analyzed patient samples. They nicely showed the effect of different methylation regions on RNA expression, and in addition in their patient samples grouped into three categories, they correlated methylation, RNA expression, and PDL1 staining.

Overall, the manuscript is well written, the methods and results are clearly presented, and the conclusion is sound.

Some minimal text editing is required

Reviewer 2 Report

Abstract should be more  concise and clear with clear and focused Abstract conclusion like, correlation or  inverse correlation of PD-L1 CpG-sites regarding methylation and their relation to the dCas9-TET1 and dCas9-DNMT3Am  to the effect of  PD-L1 hypo- and hyper-methylation

 What is correlation of high methylation and the PD L1expression  

Fig 1 needs clarification  and clear explanation of Fig Legend

Correlation is needed between  PD-L1 expression and DNA Methylation

Fig 1 CDE need to  be  rearranged as one figure

Fig 2 needs clarification PD-L1 methylation and mRNA expression

Fig 3, 4, 5 needs re-arrangement, show only significant  figure PD L1

Expression  and  Methylation

In the Introduction and Discussion elaborate  more on methylation process and its correlation  to  CpG-sites and PD L1 gene

Reviewer 3 Report

In this research article, Trine Larsen et al. experimented to address an immunotherapy issue in an epigenetic way. This article is of novelty and clinical relevance. However, this paper cannot be published without the below listed modifications:

1.      More information is required for better understanding:

·        In Fig 1,

i)       Why did authors start the mRNA and DNA methylation measurements in HCC827 and HCC827-derived cell clones, but not with the cell lines in Fig 1E? Any reasons for that? cg03379064 was brought up and denied within one figure.

ii)     Why did authors pick the unannotated CpG site cg(chr9:5,450,951)? Any reasons for that?

·        Fig 2 is quite difficult to understand,

i)       Any explanation for the difference between PD-L1 mRNA expression level and PD-L1 TPS?

ii)     ‘The significant correlation between cg19724470 245 methylation and PD-L1 TPS was largely driven by high methylation of tumors with PD-246 L1 TPS <1%.’ Where does this conclusion come from? If it is from an old publication, please specify the reference.

iii)   Are there any conclusions drew from Fig 2?

iv)    Are there different colors for the dots in Fig 2C? It is difficult to read and there are no figure legends.

2.      Resolution of images needs to be improved, the titles and annotations in several figures are hard to read.

3.      The authors included a lot of information in Supplementary Data, but the supplementary data was not properly uploaded.

4.      This article needs to be reorganized and restructured. This current version didn’t do a good enough job describing the data in a story-telling way and emphasizing the important data. For example,

·        A lot of data in supplementary seems to be more relatable than the data in main figures, but the authors didn’t upload the supplement data.

·        Cg04478497, cg00975815, and cg03379064 mentioned in Fig 1 are misleading, the article should just start with cg19724470.

·        Only until Fig 4 the conclusion was finally made about the disassociation of DNA methylation and PD-L1 mRNA level, but that’s the premise of this paper. It’s confusing to follow.

5.      In Fig 5, there was some positive data showing the impact of direct recruitment of enzymatic activities involved in methylation and demethylation to the PD-L1 gene for mRNA expression. However, no more follow-up experiments for further validation, no more discussion or conclusion making, no future directions, the article just ended in a sketchy way.

Reviewer 4 Report

* The authors investigated an essential topic regarding immunotherapy for cancer. The language of this study is not easy to understand and needs to be revised by a specialized company.

* The abstract is hard to understand and needs to be written. All abbreviations need to be defined in their first mention. 

* The introduction had sufficient information but lacked information about  Non-Small Cell 3 Lung Cancer and the mechanisms involved in its pathogenesis related to your study such as IFN.  

* Line 99: the authors mentioned the source of cell lines is ATCC. Please, mention their origin and all information related to the source.

* Line 126: please, mention the ethical approval number related to obtaining samples from patients and mention all fixation procedures of formalin and mention all steps of immunohistochemistry. 

* Line 134: please, mention the used primers.

* The supplementary files provide only the images shown in your original manuscript. Please, provide all supplementary files mentioned in the text.

* Line 235: Please, provide the immunohistochemical images in the supplementary part 

Round 2

Reviewer 3 Report

This revised version did address most of the previous issues and did a good job reorganizing all the data in a more logical, understandable way. I suggest this article be accepted.